# Nanoindentation of Graphene/Phospholipid Nanocomposite: A Molecular Dynamics Study

**DOI:** 10.3390/molecules26020346

**Published:** 2021-01-11

**Authors:** Vladislav V. Shunaev, Olga E. Glukhova

**Affiliations:** 1Department of Physics, Saratov State University, 410012 Saratov, Russia; shunaevvv@sgu.ru; 2Institute for Bionic Technologies and Engineering, I.M. Sechenov First Moscow State Medical University (Sechenov University), 119991 Moscow, Russia

**Keywords:** graphene, phospholipids, molecular dynamics, nanoindentation, local stress, electron transfer

## Abstract

Graphene and phospholipids are widely used in biosensing and drug delivery. This paper studies the mechanical and electronic properties of a composite based on two graphene flakes and dipalmitoylphosphatidylcholine (DPPC) phospholipid molecules located between them via combination of various mathematical modeling methods. Molecular dynamics simulation showed that an adhesion between bilayer graphene and DPCC increases during nanoindentation of the composite by a carbon nanotube (CNT). Herewith, the DPPC molecule located under a nanotip takes the form of graphene and is not destroyed. By the Mulliken procedure, it was shown that the phospholipid molecules act as a “buffer” of charge between two graphene sheets and CNT. The highest values of electron transfer in the graphene/DPPC system were observed at the lower indentation point, when the deflection reached its maximum value.

## 1. Introduction

Graphene is a two-dimensional allotropic modification of carbon with a thickness of one atom [1]. High biocompatibility [2], unique adsorption properties [3,4], and large surface area of graphene allow it to form a compatible interface with phospholipid molecules [5,6,7,8]. Both graphene and phospholipids separately are widely used in biosensing [9,10,11,12,13] and drug delivery [14,15,16]. It is predicted that the synergistic effect of graphene and phospholipids could be used in various biomedical devices [17,18,19,20,21].

Simulation of graphene-based composites’ nanoindentation by molecular dynamic methods allows to study local characteristics of the considered objects, as it makes it possible to place a nanotip strictly above the surface of interest. Huang et al. MD research showed that the strength and hardness of bio-inspired nanocomposites decreased with the increasing length of graphene layers, while increasing the length of the graphene layer avoids the generation of dislocations at the edge of the graphene sheet [22]. Nanoindentation of Cu/Gr layered nanopillars allowed to find the dependence of the composite mechanical properties on the specified boundary conditions and anomalous extrinsic size effect—the weakening influence caused by the dislocations at the edges of graphene was compensated by hardening of graphene in the middle of the sheet [23]. Peng et al. demonstrated that the strength of a copper substrate during indentation dramatically depended on the number of graphene layers on its surface [24]. Simulation of lipid bilayer and graphene under indentation loads has shown that a graphene coating could effectively maintain the structural and physiological stability of a bio-nanohybrid [25]. Modeling of nanoindentation showed the strengthening effect of graphene coverage on a nickel substrate [26].

The object of this study is a composite based on two graphene flakes and dipalmitoylphosphatidylcholine (DPPC) phospholipid molecules located between them. Previously, the authors found the configurations of such composites that provide optimal current–voltage characteristics and electron transfer values [27]. The aim of this work is to simulate nanoindentation of the bilayer graphene/DPPC composite with subsequent evaluation of the mechanical and electronic properties during deflection.

## 2. Results

### 2.1. Atomistic Model

At the initial stage, two DPPC molecules containing 1 phosphorus atom, 1 nitrogen atom, 8 oxygen atoms, 40 carbon atoms, and 80 hydrogen atoms were placed between two graphene monolayer flakes with dimensions L_x_ × L_y_ = 25.5 Å × 35.51 Å. This composite structure acted as the supercell with translation vectors L_x_, L_y_ (Figure 1a). This supercell contained 878 atoms, 618 of which belonged to graphene, and 260 to the DPPC molecules. The geometric center of each DPPC molecule was located at a distance of 6.23 Å from each graphene sheet. The atomic structure of the designed supercell and the values of the translation vectors were optimized by the self-consistent charge density functional tight-binding (SCC DFTB) 2 method.

The fragment of a graphene/DPPC composite film containing 7902 atoms was built from nine optimized supercells. The fragment sizes were 90.46 Å × 80.56 Å, and the initial structure is shown in Figure 1a. As the search for the ground state of such a polyatomic fragment by the SCC DFTB 2 quantum mechanical method is practically impossible, the AMBER empirical method was applied. It should be noted that the fragment is not a supercell, but a finite structure. From the result of 20 numerical experiments of the graphene/DPPC composite optimization, two types of object’s topology were identified. In the type I topology, 12 DPPC molecules formed a disordered bundle (Figure 1b). If an additional number of DPPC molecules are added to the composite, double and triple layers of lipids can be formed in this structure [5]. In the type II topology, the phospholipids were arranged in a shape vaguely resembling a spiral; part of one DPPC even left the pores between the graphene sheets (Figure 1c). During optimization, the total energy of the system decreased from 17.54, to 3.967 Mcal/mol and the adhesion energy between graphene flakes and DPPC molecules dropped from −896.115 to −4124.72 kcal/mol. Because, in this structure, two isolated DPPC molecules located in the center of the composite were clearly distinguished, it was chosen as the object of nanotip indentation.

### 2.2. Nanoindentation

At the next stage, the carbon nanotube (CNT) (16,0) with a closed edge was placed at a distance of 4.1 Å from the surface of the upper graphene sheet (Figure 2a). The length of the CNT was 50.62 Å. To perform the process of the composite material indentation, the nanotip approached the composite surface with a step of 0.5 Å. At each step, the process of relaxation scanning was started, the values of local stresses on the atoms were determined, and the adhesion energy between bilayer graphene and phospholipids was calculated. Based on the analysis of the above-mentioned characteristics, the so-called “key points” of indentation were identified (Table 1).

The dependence of the system’s total energy on the CNT shift is shown in Figure 2c. A local minimum was observed at point A, indicating the most energetically favorable location between the CNT and the composite owing to the van der Waals (vdW) interaction. Further, the energy grew steadily until it reached point C. Between points C and D, there was a slight drop in the total energy of the system. In this region, the atomic structure of the DPPC molecule was rearranged and it started to take the form of graphene (Figure 2b). Note that, during the forward stroke, the adhesion energy between graphene and DPPC molecules reached its maximum value precisely at point D (Table 1). The position of CNTs at the lower indentation point F is shown in Figure 2a on the right. As the CNT left the trough of the composite (reverse stroke), the energy decreased greatly (from F to H). At this interval, the CNT tip stopped to pressurize the composite and began to relax. The point H corresponds to the moment of the strongest vdW interaction between objects during reverse stroke, so the local minimum of the total energy at this point is observed. Then, the vdW interaction became weaker and, from a certain moment (point I), the total energy practically did not change. Note that the trough created by the nanotip in the composite remained even after the CNT returned to its initial position. Herewith, the remaining 16 DPPC molecules did not leave the space between the graphene sheets (Figure 2a). The value of the adhesion energy between graphene and DPPC molecules at point I and further remained at the maximum for the entire time. Thus, despite the fact that no chemical bonds between graphene and DPPC molecules were formed during indentation, the bonds between graphene sheets and DPPC molecules were significantly strengthened.

### 2.3. Analysis of Graphene Sheets Strength during Nanoindentation

To assess the strength of graphene sheets during deflection, we calculated the map of the local stresses’ distribution by atoms at each step (Figure 3a–h). As the maximum values of local stresses (MLSs) were found in the central atoms of the sheets (under nanotip), we presented maps only for these regions. Figure 3i shows the graph of the MLS on graphene atoms dependence on the indentation step. From the beginning of indentation to point B, the MLSs were observed in the upper sheet of graphene and varied from 0 to 0.47 GPa. Such minor changes were caused by the fact that the CNT has not yet reached the surface of the upper graphene sheet. The dependence on the BE section was almost linear, which indicated the elastic nature of the deformation. In this segment, the central phospholipid molecule under the CNT has taken the form of the curved graphene sheet. At the same time, there was significant increase in the MLS from 0.46 to 2.25 GPa. Therefore, at the BE segment, the graphene sheets started to provide a strengthening effect on the DPPC molecule, not allowing it to destruct under the influence of the nanotip. After this point, the adhesion between graphene and DPPC increased (see Section 2.2) and the MLS did not change much; on the EF segment, the MLS increased from 2.25 to 2.53 GPA. During the reverse stroke, a sharp drop in the MLS from 2.53 to 0.86 GPA was observed between F and G. At this interval, the composite started to relax because pressure from the tip became weaker (see Section 2.2). Starting from point G, the MLSs were observed in the lower graphene sheet and, starting from point H, the MLS values stopped changing because the energy of the vdW interaction between objects reached a minimum.

### 2.4. Analysis of Electron Transfer in the CNT/Graphene/DPPC System during Nanoindentation

As the calculation of atoms’ Mulliken charges by the SCC DFTB 2 method is resource consuming, the central part was cut out from the graphene/DPPC composite. This fragment containing two isolated DPPC molecules directly contacted the CNT and was responsible for charge transfer in the system (Figure 4a).

The charge distributions between the atoms at the initial moment of time, at the lower point of the forward stroke F, and at the end of the reverse stroke I are shown in Table 2 At the initial moment of time, the CNT was electrically neutral. In the composite, the DPPC molecules acted as a donor and transferred the charge of 6.09 e to graphene sheets; the largest part of the charge was lost by two P-atoms (1.61 and 1.65 e), which is consistent with the results obtained earlier [26]. The charge between the graphene sheets was distributed unevenly because of the orientation of the DPPC molecules after optimization—the phosphorus molecules that tend to give charge and act as donors were located closer to the upper graphene sheet. During the indentation, the CNT gradually transferred the charge to the composite; at the lower point, its value was 0.16 e (Figure 4c). It can be seen that the phospholipid transferred even more charge to graphene sheets than in the initial state (7.28 e) (Figure 4d). The charge transferred from DPPC was evenly distributed between the graphene sheets (Figure 4b). This was caused by the fact that the DPPC molecules have taken the form of graphene sheets, as shown in Section 2.2, and the P atoms were located at the same distance from the graphene sheets. As the CNT is removed from the composite, it recovered the transferred charge and eventually became almost electroneutral again. Note that, during the reverse stroke, the main part of the charge came from the upper sheet of graphene that was directly in contact with the CNT. Thus, the DPPC molecules stopped to act as a “buffer” of charge between the two graphene sheets. The highest values of electron transfer in the graphene/DPPC system were observed at the lower indentation point, when the deflection reached its maximum value. Based on the conclusions of [26], it can be concluded that the deflection strain would significantly affect the current–voltage characteristics of the considered composite.

## 3. Methods

The search for the ground state of the graphene/DPPC composite film, as well as the study of its atomic structure changes during deflection by a nanotip, was performed by the AMBER empirical method [28] implemented in the Hyperchem software package [29]. Optimization was performed by the conjugate gradient Fletcher–Reeves method, and the root mean square (RMS) gradient was 0.1 kcal/(A·mol).

The adhesion energy between the phospholipids and graphene sheets at various steps was calculated by the following formula (Equation (1)):(1)$$E_{ADH}=E_{TOT}-\;E_{DPPC}-E_{UP}-E_{LOW}\;\;$$
where $E_{\mathrm{TOT}}$ is the energy of the graphene/DPPC system at this point, and $E_{\mathrm{DPPC}}$, $E_{\mathrm{UP}}$, and $E_{\mathrm{LOW}}$ are the energies of isolated DPPC, upper, and lower graphene sheets, respectively.

To estimate the strength of graphene sheets during deflection, the previously developed original method for calculating local stresses of the atomic grid was used [30]. According to this method, the stress *σ_i_* on each atom is calculated by the formula $\sigma_{i}=\left|w_{i}-w_{0}\right|$, where $w_{0}$ is the energy volume density of the graphene atom before indentation, and $w_{i}$ is the energy volume density of the graphene atom under external influence. The energy volume density of the atom was calculated by the formula wi=EiVi, where $E_{i}$ is the energy of the atom calculated within the AMBER force field, and Vi=4πr33 is the volume of the carbon atom (*r* = 1.7 Å).

The study of changes in the electronic structure during indentation and electronic transfer between phospholipid molecules and graphene was performed by Mulliken population analysis [31]. According to this method, the charge on an atom is calculated as the difference between the atomic number $Z_{A}$ and $GAP_{A}$—the sum of the gross orbital product over all orbitals belonging to atom *A*: $Z=\;Z_{A}-GAP_{A}$. The charges were calculated by the quantum mechanical self-consistent charge density functional tight-binding (SCC DFTB) method [32] in the dftb+ software package [33] in the 3ob-3-1 basis.

## 4. Conclusions

The nanoindentation of the composite on the base of bilayer graphene and 16 DPPC phospholipid molecules was simulated by the molecular dynamics method. It was noted that, during indentation, the adhesion between graphene flakes and DPPC molecules increased and reached a maximum at the end of the reverse stroke. The maximum values of local stresses (in the region of 2.53 GPA) were observed on the upper graphene layer at the lower indentation point. At this moment, the DPPC molecule located under the nanotip took the form of curved graphene, while chemical bonds of the phospholipid molecules were not destroyed. The pressure of the CNT tip led to the growth of adhesion energy between graphene sheets and DPPC molecules. It is known that, getting into blood vessels, drug carriers sense abnormally high shear stresses [34,35], so the discovered effect of “phospholipid strengthening” by the graphene sheets’ coating can be used in the field of drug delivery. It was found that the electron transfer between CNT and graphene/DPPC composites increased during indentation and reached 0.16 e. At this moment, the phospholipid molecule stopped to act as a “buffer” of charge between the two graphene sheets. The observed phenomenon of electronic transfer between graphene and phospholipid can be applied in biosensorics.

## Figures and Tables

**Figure 1 molecules-26-00346-f001:**
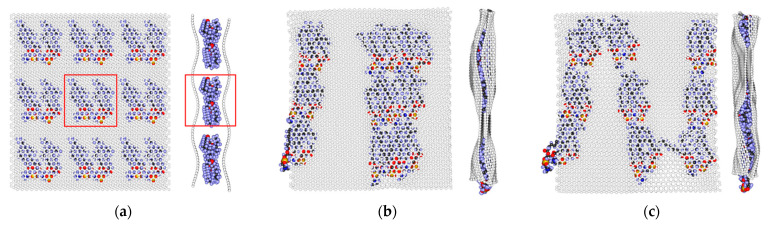
Atomic structure of the graphene/dipalmitoylphosphatidylcholine (DPPC) composite: (**a**) before the optimization; (**b**) type I after the optimization; and (**c**) type II after the optimization. The supercell obtained by the self-consistent charge density functional tight-binding (SCC DFTB) 2 method is highlighted by the red box. Graphene atoms are grey, phosphorous—yellow, nitrogen—blue, oxygen—red, carbon in phospholipid—black, and hydrogen—blue.

**Figure 2 molecules-26-00346-f002:**
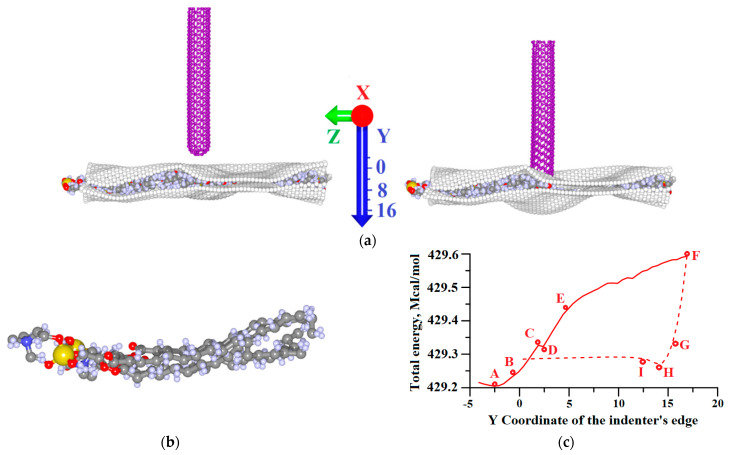
Nanoindentation of the composite graphene/DPPC by the carbon nanotube (CNT) with a closed edge: (**a**) initial (left) and last (right) point of indentation F; (**b**) the atomic structure of the central DPPC molecule at point F; and (**c**) the dependence of the system’s total energy on the CNT shift. The solid line corresponds to the forward stroke (FS) and the dotted line corresponds to the reverse. At the initial point, the edge of the CNT had the coordinate Y = 0, and the atoms of the upper graphene layer had the coordinate Y = −4.1 Å.

**Figure 3 molecules-26-00346-f003:**
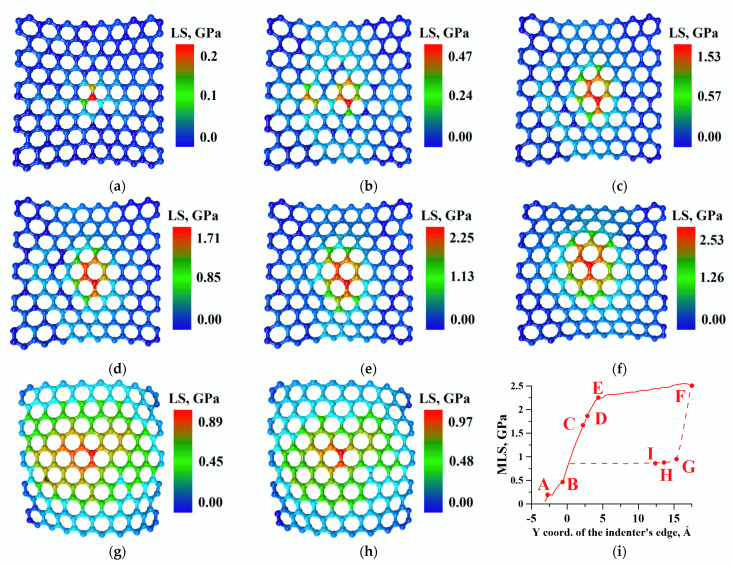
The map of the local stress (LS) in the central area of the graphene sheets at the key points of indentation: (**a**) point A; (**b**) point B (upper layer); (**c**) point C (upper layer); (**d**) point D (upper layer); (**e**) point E (upper layer); (**f**) point F (upper layer); (**g**) point G (lower layer); (**h**) point H (lower layer); and (**i**) the dependence of the maximum local stress (MLS) in graphene sheet atoms on the indentation step (the solid line corresponds to the forward stroke and the dotted line corresponds to the reverse).

**Figure 4 molecules-26-00346-f004:**
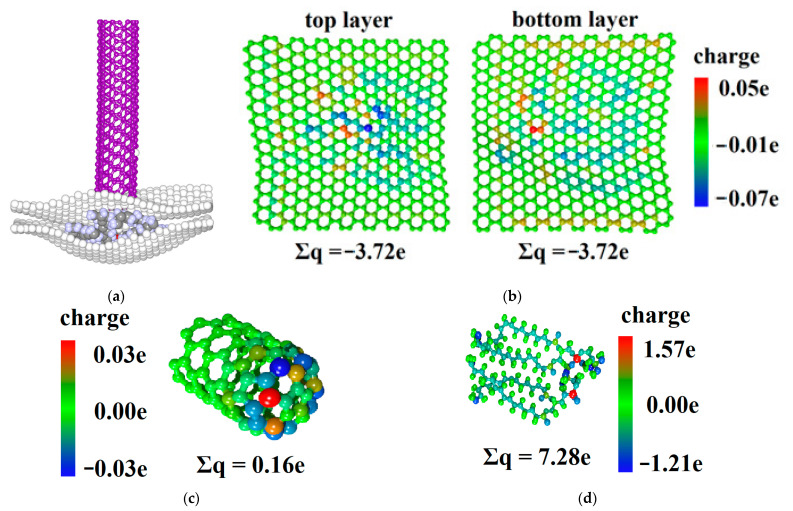
The analysis of the electron transfer in the CNT/graphene/DPPC system during nanoindentation (**a**) The fragment of the atomic structure graphene/DPPC with CNT at point F; (**b**) the distribution of Mulliken charges on atoms in the point for the upper (left) and the lower (right) graphene sheets; (**c**) for CNT; and (**d**) for DPPC.

**Table 1 molecules-26-00346-t001:** Key points of the nanoindentation with corresponding Y coordinates; the values of the adhesion energy between bilayer graphene and dipalmitoylphosphatidylcholine (DPPC) molecules and maximum local stresses (MLSs) in these points. The abbreviation FS corresponds to forward stroke and RS corresponds to reverse stroke.

Key Point	Y Coordinate of the Indenter’s Edge, Å	Energy of Adhesion between Bilayer Graphene and Phospholipid, kcal/mol	MLS, GPa
**A**	−2.6 (FS)	−4125.1	0.2
**B**	−0.6 (FS)	−4124.03	0.47
**C**	1.9 (FS)	−4125.85	1.53
**D**	2.4 (FS)	−4127.13	1.71
**E**	4.4 (FS)	−4114.53	2.25
**F**	17.4(FS)	−4116.33	2.53
**G**	15.4 (RS)	−4166.4	0.89
**H**	13.4 (RS)	−4172.05	0.97
**I**	12.4 (RS)	−4172.45	0.97

**Table 2 molecules-26-00346-t002:** Distribution of Mulliken charge in the system CNT/graphene/DPCC at different moments of nanoindentation. CNT, carbon nanotube.

	CNT	Upper Layer of Graphene Sheet	DPPC	Lower Layer of Graphene Sheet
**Initial moment of time**	0.00 e	−3.45 e	6.09 e	−2.63 e
**Point F**	0.16 e	−3.72 e	7.28 e	−3.72 e
**Point I**	−0.01 e	−3.28 e	6.87 e	−3.58 e

## Data Availability

The data presented in this study is available in this article.

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
