# Peer review of "Nanoindentation of Graphene/Phospholipid Nanocomposite: A Molecular Dynamics Study"

_molecules, 2021, doi:10.3390/molecules26020346_

Round 1

Reviewer 1 Report

The authors present MD simulation of nanoindentation of a graphene/DPPC sheet by a CNT tip.  The simulation method is sound and the results are described reasonably well.  The main drawback is the quality of English.  I've indicated the required changes in the attached commented pdf.

Author Response

The authors of the article thank the reviewer for his kind help in improving English quality of the paper. All required changes were made and highlighted by yellow.

Reviewer 2 Report

The paper titled by “Nanoindentation of Graphene/Phospholipid Nanocomposite: a Molecular Dynamics Study”, reports a systematic study of the mechanical and electronic properties of the configuration including 16 dipalmitoylphosphatidylcholine (DPPC) phospholipid molecules located between two graphene flakes. The nanoindentation of graphene/phospholipid nanocomposite by attaching a carbon nanotube was simulated by molecular dynamics.  The structures of nanocomposites before and nanoindentation were reported. Moreover, the stress and electron transfer behavior were obtained during the nanoindentation process. As a result, the discovered effect of "phospholipid strengthening" by graphene sheets coating, as well as the phenomenon of electronic transfer between graphene and phospholipid can be used in drug delivery and biosensorics. Many persuasive theoretical calculations have been carried out to support the authors’ conclusions. Overall, I rate this manuscript as scientifically sound, and interesting for the audience of molecules. However, some minor revisions are required to add to the value of the manuscript for definitely supporting its acceptation.

  • A very general comments: please explain the biotech meaning more clearly for audience from different background that why the nanoindentation performance of your system is important for drug delivery? How the mechanical stability and electronic transfer properties contribute to its application in drug delivery? Otherwise, there is a big gap for understanding between your basic findings in this paper and its promising application in drug delivery you supposed.

  • As for Figure 2(c), please explain clearly about the reverse stroke in the manuscript. Why the total energy is greatly decreased from F to H, then increased a little bit again from H to I? This part of performance was ignored in the discussion of section 3.2.    

  • As for Figure 3(i), please explain the reason for why you observe this kind of MLS performance from E to I. For example, why does it keep steady from E to F and after G, also why does it greatly decrease from F to G?

  • In conclusion section: “To this moment, the DPPC molecule located under the nanotip took the form of curved graphene and was not destroyed.” What do you mean by “not destroyed”? In comparison with the sentence (line 120, page 4) “Note that the trough created by the nanotip in the composite remained even after the CNT returned to its initial position.” Is there any conflicts between the two sentences?  Is this performance of the trough remaining there after nanoindentation good for the application in drug delivery? Why?  

Author Response

We thank reviewer for his questions and remarks. The made changes are highlighted in the text by blue.

  • A very general comments: please explain the biotech meaning more clearly for audience from different background that why the nanoindentation performance of your system is important for drug delivery? How the mechanical stability and electronic transfer properties contribute to its application in drug delivery? Otherwise, there is a big gap for understanding between your basic findings in this paper and its promising application in drug delivery you supposed.

The following text was added to the paper (lines 187-191, page 7)

The pressure of CNT tip lead to the growth of adhesion energy between graphene sheets and DPPC molecules. It’s known that getting into blood vessels, drug carriers sense abnormally high shear stresses [34, 35], so the discovered effect of "phospholipid strengthening" by graphene sheets coating can be used in the field of drug delivery

The papers 34-35 were added to References

  1. Xia Y., Shi C.-Y., Xiong W. et al. Shear Stress-sensitive Carriers for Localized Drug Delivery. Current Pharmaceutical Design 2016, 22(38), 5855-5867.
  2. Godoy-Gallardo M., Ek M. P., Jansman M. M. T. et al. Shear Stress-sensitive Carriers for Localized Drug Delivery. Biomicrofluidics 2015, 9,

As for Figure 2(c), please explain clearly about the reverse stroke in the manuscript. Why the total energy is greatly decreased from F to H, then increased a little bit again from H to I? This part of performance was ignored in the discussion of section 3.2.   

The following text was added to the paper (lines 121-125, page 4)

As the CNT left the trough of the composite (reverse stroke), the energy greatly decreased (from F to H). At this interval CNT tip stopped to pressurize the composite and it began to relax. The point H corresponds to the moment of the strongest vdW interaction between objects during reverse stroke, so the local minimum of the total energy at this point is observed. Then the vdW interaction became weaker and from a certain moment (point I) the total energy practically didn’t change.

  • As for Figure 3(i), please explain the reason for why you observe this kind of MLS performance from E to I. For example, why does it keep steady from E to F and after G, also why does it greatly decrease from F to G?

The following text was added to the paper (lines 144-150, page 4)

After this point, the adhesion between graphene and DPPC increased (see Section 3.2) and the MLS didn’t change much: on the EF segment the MLS increased from 2.25 to 2.53 GPA. During the reverse stroke the sharp drop in the MLS from 2.53 to 0.86 GPA was observed between F and G. At this interval the composite started to relax since pressure from the tip became weaker (see Section 3.2). Starting from point G, the MLS were observed in the lower graphene sheet, and starting from point H, the MLS values stopped changing since the energy of the vdW interaction between objects became minimum.

  • In conclusion section: “To this moment, the DPPC molecule located under the nanotip took the form of curved graphene and was not destroyed.” What do you mean by “not destroyed”? In comparison with the sentence (line 120, page 4) “Note that the trough created by the nanotip in the composite remained even after the CNT returned to its initial position.” Is there any conflicts between the two sentences?  Is this performance of the trough remaining there after nanoindentation good for the application in drug delivery? Why?  

There is no conflict between to sentence. DPPC molecules deformed and took the form of the graphene, but its chemical bonds were not destroyed (this clarification was added to the text). This fact confirms strengthening effect of graphene.

The following text was added to the paper (lines 187-191, page 7)

It’s known that getting into blood vessels, drug carriers sense abnormally high shear stresses [34, 35], so the discovered effect of "phospholipid strengthening" by graphene sheets coating can be used in the field of drug delivery

The papers 34-35 were added to References

  1. Xia Y., Shi C.-Y., Xiong W. et al. Shear Stress-sensitive Carriers for Localized Drug Delivery. Current Pharmaceutical Design 2016, 22(38), 5855-5867.
  2. Godoy-Gallardo M., Ek M. P., Jansman M. M. T. et al. Shear Stress-sensitive Carriers for Localized Drug Delivery. Biomicrofluidics 2015, 9,
